# Depletion of Small HDL Subclasses Predicts Poor Survival in Liver Cirrhosis

**DOI:** 10.3390/antiox14060664

**Published:** 2025-05-30

**Authors:** Anja Pammer, Tobias Madl, Hansjörg Habisch, Jakob Kerbl-Knapp, Florian Rainer, Vanessa Stadlbauer, Angela Horvath, Philipp Douschan, Rudolf E. Stauber, Gunther Marsche

**Affiliations:** 1Division of Pharmacology, Otto Loewi Research Center, Medical University of Graz, 8010 Graz, Austria; anja.pammer@medunigraz.at; 2Medicinal Chemistry, Otto Loewi Research Center, Medical University of Graz, 8010 Graz, Austria; tobias.madl@medunigraz.at; 3BioTechMed-Graz, 8010 Graz, Austria; hansjoerg.habisch@medunigraz.at; 4Division of Molecular Biology and Biochemistry, Gottfried Schatz Research Center, Medical University of Graz, 8010 Graz, Austria; jakob.kerbl-knapp@unibas.ch; 5Division of Gastroenterology and Hepatology, Department of Internal Medicine, Medical University of Graz, 8036 Graz, Austria; florian.rainer@medunigraz.at (F.R.); vanessa.stadlbauer@medunigraz.at (V.S.); angela.horvath@medunigraz.at (A.H.); philipp.douschan@medunigraz.at (P.D.); rudolf.stauber@medunigraz.at (R.E.S.); 6Center for Biomarker Research in Medicine (CBmed), 8010 Graz, Austria; 7Division of Pulmonology, Lung Research Cluster, Medical University of Graz, 8036 Graz, Austria

**Keywords:** HDL subclasses, oxidative stress marker, NMR, liver failure, cirrhosis, mortality

## Abstract

Liver cirrhosis is a complex condition characterized by oxidative stress, inflammation, and metabolic dysfunction, contributing to systemic complications and high mortality. High-density lipoprotein (HDL), particularly its small subclasses, is known for its antioxidant, anti-inflammatory, and cholesterol efflux capacities. This study examined changes in HDL subclass distribution and composition in cirrhosis and assessed their prognostic relevance for mortality. We analyzed HDL profiles using nuclear magnetic resonance spectroscopy in patients with compensated (n = 205) and decompensated (n = 158) cirrhosis, compared to healthy controls (n = 16). Across all HDL subclasses in cirrhotic patients, cholesterol content was decreased, and triglyceride levels were elevated. In particular, compensated cirrhosis was associated with a marked reduction in small and extra-small HDL particles, while large HDL levels remained unchanged. This reduction was even more pronounced in decompensated disease. Small HDL particle levels were inversely correlated with oxidative stress markers and liver dysfunction and independently predicted 12-month mortality in patients with compensated cirrhosis, even after adjusting for MELD score. In conclusion, our findings highlight a substantial depletion of small and extra-small HDL particles as a key feature of cirrhosis, linked to oxidative stress and mortality in the compensated stage.

## 1. Introduction

Chronic liver disease constitutes a significant global health challenge, accounting for approximately two million deaths annually [1]. Emerging evidence highlights the pivotal role of dysregulated lipid metabolism in orchestrating inflammatory pathways and exacerbating disease severity [2,3,4]. In particular, decreased levels of high-density lipoprotein (HDL) cholesterol and its primary apolipoprotein, apolipoprotein A-I (ApoA-I), have been consistently associated with heightened systemic inflammation, greater disease severity, and reduced survival in individuals with cirrhosis [5].

Multiple mechanistic pathways support the protective role of HDL in liver disease. First, HDL exhibits antioxidative properties that counteract oxidative stress, a central driver of hepatocellular injury and vascular remodeling [6,7]. Second, HDL participates in reverse cholesterol transport, thereby reducing lipid accumulation contributing to hepatic damage [8]. Third, in cirrhotic patients, impaired hepatic function and portal hypertension lead to increased intestinal permeability, facilitating the translocation of microbial products, such as lipopolysaccharides (LPS), into the systemic circulation. HDL may mitigate these effects by neutralizing circulating endotoxins and promoting tissue repair, thus potentially attenuating disease progression. These multifaceted functions position HDL as a promising therapeutic target for the management of cirrhosis-related complications [4,9,10]. However, the reduced concentrations and impaired functionality of HDL in cirrhosis are likely to compromise these protective mechanisms, thereby exacerbating systemic inflammation and underscoring the dual role of HDL as a biomarker and potential therapeutic target. The functional heterogeneity of HDL is reflected in its diverse subclasses, which differ in size, density, and compositional characteristics. Among these, small, dense HDL subpopulations—characterized by distinct protein and lipid profiles—have attracted particular clinical interest due to their pronounced atheroprotective and anti-inflammatory properties. These subclasses exhibit enhanced cholesterol efflux capacity, potent antioxidant and anti-inflammatory effects, and a superior ability to inhibit endothelial cell apoptosis [7,11]. Notably, recent evidence suggests that these small HDL subclasses offer improved prognostic value in assessing mortality risk in various pathological conditions, surpassing the predictive accuracy of conventional total HDL cholesterol measurements [12,13,14,15]. In the present study, we used proton nuclear magnetic resonance (NMR) spectroscopy to analyze the distribution and compositional changes in HDL subclasses in patients with cirrhosis. Our study investigates the link between small HDL subclasses and markers of oxidative stress and liver dysfunction and whether specific subclass shifts are associated with mortality.

## 2. Materials and Methods

### 2.1. Study Cohort

The plasma samples of the study cohort were obtained from a previously published study of 508 patients with cirrhosis that investigated HDL-related biomarkers in chronic liver failure [5]. Of these, 363 plasma samples were available for further analysis. The current study included two distinct groups: (i) 205 consecutive patients with stable cirrhosis recruited between 2011 and 2016 at the Medical University of Graz (from the Hepatology Outpatient Clinic or the Gastroenterology/Hepatology Ward, including stable patients undergoing evaluation for liver transplantation), and (ii) 158 patients with decompensated cirrhosis, with or without acute-on-chronic liver failure (ACLF), enrolled between February and September 2011 as part of the multicenter CANONIC study in 12 European countries (see Moreau et al., 2013 [16] for details). In both cohorts, cirrhosis was diagnosed based on liver histology or a combination of clinical, biochemical, and imaging criteria. Patients with a history of solid organ transplantation or hepatocellular carcinoma with advanced Barcelona Clinic Liver Cancer stages C and D were excluded. In addition, individuals with cholestatic liver disease were not included because of the effect of cholestasis on lipid profiles. Hospitalized patients with cirrhosis were evaluated for acute decompensation or ACLF, as defined by Moreau et al. in 2013 [16]. In addition to the collection of baseline data, including medical history, physical examination, and laboratory measurements, information on liver transplantation and mortality at 90 days and 12 months after the start of the study was collected.

Additionally, 16 age- and sex-matched healthy controls, who did not meet the following exclusion criteria, were included: any history of cardiovascular disease, pregnancy, obesity, dyslipidemia, liver disease, renal disease, diabetes, or clinical signs of inflammation. The control participants were not taking any medication that lowers cholesterol or reduces inflammation.

Blood samples were obtained from patients and healthy controls at the outset of the study. The study was approved by the local Institutional Review Board (Medical University of Graz, 23-056 ex 10/11, 23-096 ex 10/11, 23-285 ex 10/11) in accordance with the Declaration of Helsinki. Each patient was required to provide written informed consent unless the requirement for this had been waived by the local Institutional Review Board.

### 2.2. NMR Spectroscopy Measurements

Serum levels of HDL-ApoA-I levels within each subclass were quantified using a Bruker 600 MHz Avance Neo NMR (Bruker, Rheinstetten, Germany) spectrometer and are reported in mg/dL, reflecting the mass concentration of apolipoprotein A-I in plasma. These values represent protein mass associated with each HDL subclass, not particle number. NMR spectra were recorded at a constant temperature of 310 K using various pulse sequences for proton spectra acquisition and water suppression. ApoA-I is the primary structural protein of HDL, and its subclass-specific distribution provides an indirect measure of HDL particle remodeling and potential functional capacity. The Bruker IVDr lipoprotein subclass analysis protocol (B.I.LISATM) was used for assessing subclass concentrations [17].

### 2.3. Statistical Analysis

Statistical analyses were conducted using SPSS (Version 29.0.0.0) (SPSS, Inc., Chicago, IL, USA) and GraphPad Prism 10.4.1. A *p*-value of less than 0.05 was considered statistically significant. Participant characteristics are represented as the median and interquartile range (Q1–Q3) or count and proportion. Mann–Whitney U Test or Fisher’s Exact Test were used to examine differences in clinical and laboratory characteristics and HDL subclass distribution between individuals with compensated and decompensated cirrhosis. A post hoc power analysis was conducted to assess the adequacy of the control group size (n = 16) in detecting differences in HDL parameters. Using α = 0.05 and assuming a two-sided *t*-test, comparisons with patient groups (n = 205 or n = 158) showed statistical power ranging from 0.76 to 0.87 to detect effect sizes of 0.7–0.8 (Cohen’s d), which correspond to the magnitude of group differences observed in key HDL metrics. Statistical significance between healthy controls and individuals with cirrhosis was assessed using the Kruskal–Wallis test with Dunn’s multiple comparisons post hoc analysis. The associations between HDL subclasses and inflammatory/oxidative markers were evaluated using Spearman’s correlation. The prognostic value of HDL parameters for 90-day or 12-month mortality was examined using multivariable Cox regression analyses, as well as receiver operating characteristic (ROC) analysis.

## 3. Results

### 3.1. Baseline Characteristics

In this study, we analyzed 363 plasma samples from two distinct patient cohorts: 205 individuals with compensated cirrhosis from a single-center cohort in Austria and 158 individuals with decompensated cirrhosis from a European multicenter cohort, including 41 cases of ACLF. Additionally, 16 age- and sex-matched healthy controls were included for comparison (Table 1). The mean age across both cirrhosis cohorts was 58 years. Among individuals with decompensated cirrhosis, 38.6% were female, compared to 25.4% in the compensated cirrhosis group. Compared to patients with compensated cirrhosis, patients with decompensated cirrhosis had a significantly different clinical profile, characterized by lower serum albumin (*p* < 0.001) levels and higher bilirubin (*p* < 0.001) and creatinine (*p* = 0.004) concentrations. Inflammatory markers, including white blood cell count (*p* = 0.007) and C-reactive protein (*p* < 0.001), were markedly elevated in the decompensated cirrhosis group, while lipid parameters, specifically total cholesterol and HDL cholesterol, were significantly lower (*p* < 0.001). The primary cause of cirrhosis differed between cohorts: alcohol-related cirrhosis was most common in the compensated group (55.4%), whereas the decompensated group showed a more even distribution between alcohol-related (39.3%) and viral (32.3%) etiologies. Mortality rates also varied substantially, with 90-day mortality at 3.9% for patients with compensated cirrhosis and 20.3% for those with decompensated cirrhosis, while 12-month mortality rates were 7.3% and 34.8%, respectively (Table 1).

### 3.2. HDL Subclass Distribution in Cirrhosis

Proton NMR spectroscopy was used to analyze HDL subclass composition, encompassing a range from large to extra-small particles. In patients with compensated cirrhosis, a significant reduction in small to extra-small HDL (S-HD to XS-HDL) particle concentrations was observed (*p* < 0.001), with no significant change in large HDL (L-HDL) (*p* = 0.404) and medium-sized HDL (M-HDL) (*p* = 0.095). Decompensated cirrhosis exhibited a more pronounced reduction in M-HDL, S-HDL and XS-HDL subclasses (*p* < 0.001) (Figure 1).

### 3.3. Composition of HDL Subclasses in Patients with Cirrhosis

The distribution of HDL subclasses in patients with compensated and decompensated cirrhosis differed markedly from that in healthy controls, prompting a detailed analysis of HDL particle composition. Lipid levels were normalized to ApoA-I subclass concentrations to evaluate the lipid content of L- to XS-HDL particles (Figure 2). This analysis revealed significantly reduced HDL-cholesterol levels in M-HDL, S-HDL and XS-HDL subclass concentrations in all patients with cirrhosis (*p* < 0.001), (Figure 2A), alongside a notable decrease in free cholesterol content within L-HDL particles (*p* < 0.001) (Figure 2B). Phospholipid levels showed only subtle variations, with a modest increase observed in S-HDL particles (*p* = 0.015) among individuals with decompensated cirrhosis. In contrast, triglyceride concentrations were elevated across all HDL subclasses in patients with compensated and decompensated cirrhosis (*p* < 0.001) (Figure 2D).

### 3.4. Inflammation, Etiology of Liver Failure, and HDL Subclass Distribution

We examined the association between inflammation (C-reactive protein levels) and HDL subclass distribution in cirrhosis. In patients with compensated cirrhosis, elevated CRP ≥ 5 mg/L was strongly linked to a significant reduction in M-HDL (*p* = 0.038), S-HDL (*p* < 0.001), and XS-HDL (*p* = 0.013) subclass concentrations, while L-HDL (*p* = 0.052) concentrations showed only a non-significant trend. Conversely, in patients with decompensated cirrhosis, HDL subclass distribution exhibited no association with inflammation status (all *p* > 0.999) (Figure 3). Irrespective of underlying etiology, encompassing predominant causes such as alcohol-related liver disease and viral hepatitis, compensated liver cirrhosis was associated with a moderate reduction in M-HDL, S-HDL and XS-HDL subclasses (Appendix A). This observation remained consistent across both alcohol-related and hepatitis-related cirrhosis when compared to other etiologies. In patients with decompensated cirrhosis, the distribution of HDL subclasses was not significantly influenced by disease etiology (Appendix A).

### 3.5. Associations of HDL Subclasses with Markers of Oxidative Stress, Inflammation and Liver Dysfunction

A robust inverse correlation was observed between HDL subclasses and C-reactive protein (CRP) levels. The strongest association was noted for XS-HDL particles (see Figure 4). This suggests that XS-HDL particles may be highly responsive to systemic inflammation. Additionally, bilirubin exhibited a significant negative correlation with M- and S-HDL particles, indicating a potential link between HDL subclass distribution and liver dysfunction. Several amino acids exhibited inverse correlations with all HDL subclasses. Tyrosine and phenylalanine, which are aromatic amino acids that are often elevated during metabolic stress, showed particularly strong negative associations with smaller HDL particles. XS-HDL particles demonstrated the most significant negative correlations with glutamine and glycine, which are key glutathione synthesis precursors, as well as with pyruvic acid, a metabolite closely involved in redox homeostasis and antioxidant defense. Together, these results underscore the strong relationships between HDL subclasses and indicators of inflammation, liver dysfunction, and oxidative stress.

### 3.6. HDL Subclasses and Mortality in Compensated and Decompensated Cirrhosis

Analysis of HDL subclasses demonstrated a strong association with mortality. In patients with compensated cirrhosis, a significant reduction across all HDL subclasses was closely linked to an elevated 12-month mortality risk (all *p* < 0.001) (Figure 5A). Likewise, in patients with decompensated cirrhosis, lower HDL subclass concentrations were associated with increased 90-day mortality (L-HDL: *p* = 0.012, M-HDL: *p* < 0.001, S-HDL: *p* = 0.134, XS-HDL: *p* = 0.002) (Figure 5B). However, this association weakened at 12 months in the decompensated group (L-HDL: *p* = 0.024, M-HDL: *p* = 0.011, S-HDL: *p* = 0.277, XS-HDL: *p* = 0.383) (Appendix A).

### 3.7. HDL Subclasses as Predictors of Mortality in Liver Cirrhosis

To assess the independent prognostic value of HDL subclass parameters for mortality risk in liver cirrhosis, we performed multivariable Cox regression analyses (Table 2). In patients with compensated cirrhosis, after adjusting for age, sex, and Model for End-Stage Liver Disease (MELD) score, M-HDL (*p* < 0.001), S-HDL (*p* < 0.001) and XS-HDL (*p* = 0.001) showed a significant inverse association with 12-month mortality. In contrast, among patients with decompensated cirrhosis, only XS-HDL retained a significant inverse association with 90-day mortality (*p* = 0.004). Following additional adjustment for C-reactive protein (CRP) levels, the inverse association between HDL subclasses and three-month mortality was attenuated in compensated patients, with XS-HDL particles no longer reaching statistical significance (*p* = 0.246). Conversely, in cases of decompensated patients, the correlation with XS-HDL particles exhibited a strengthening *p*-value (Appendix A, *p* = 0.001).

### 3.8. Receiver Operating Characteristic Analyses of HDL Subclasses as Mortality Predictors

To assess clinical utility, receiver operating characteristic (ROC) analyses were performed on significant Cox regression predictors. In compensated cirrhosis, M-HDL had a numerically higher area under the curve (AUC) than MELD for predicting 12-month mortality (0.92 vs. 0.89; Figure 6A). The combination of M-HDL-ApoA-I and MELD further improved prognostic accuracy (AUC: 0.93). In decompensated cirrhosis, although XS-HDL-ApoA-I was a significant predictor in Cox regression, it did not significantly improve the 90-day mortality prediction of MELD (Figure 6B). Although the ROC curves of combined MELD and M- HDL or combined MELD and XS-HDL showed comparable or slightly higher AUC than the MELD score alone, statistical comparison using the DeLong test revealed no significant differences between the AUCs of the MELD score and those combined parameters (compensated: *p* = 0.612, decompensated: *p* = 0.818).

## 4. Discussion

Cirrhosis markedly alters lipoprotein metabolism, leading to substantially reduced HDL and total cholesterol levels. This study offers the first comprehensive analysis of HDL subclass composition in cirrhosis and its clinical implications for mortality. We observed a significant decline in M-HDL to XS-HDL particle concentrations in patients with cirrhosis, particularly those with decompensated disease, whereas L-HDL levels remained relatively unchanged. This decrease was associated with lower cholesterol content within the M-to-XS-HDL, whereas triglyceride levels were increased in all subclasses. Notably, reduced M-to-XS-HDL concentrations were independently linked to higher mortality risk, even after adjusting for established predictors such as age, sex, and MELD score, underscoring their potential as a prognostic marker.

Further adjustment for CRP levels (an indicator of systemic inflammation), revealed nuanced insights. For compensated patients, the protective association between HDL subclasses and three-month mortality weakened, and the effect of XS-HDL particles lost statistical significance. Conversely, for decompensated patients, the association with XS-HDL particles persisted and strengthened. These findings suggest that, for compensated patients, the relationship between HDL subclasses and mortality may be partially influenced by inflammation. For decompensated patients, XS-HDL particles may serve as more robust and independent prognostic markers.

These findings are consistent with the recognized biological roles of HDL subclasses, wherein critical functions—such as cholesterol efflux capacity, anti-oxidative capacity, anti-inflammatory effects, and antiapoptotic activity—are predominantly mediated by small, dense, protein-rich HDL particles [11]. Specifically, these smaller HDL particles may confer significant protection against oxidative stress induced by free radicals [11].

Our study shows a significant inverse correlation between circulating levels of small HDL subclasses and key markers of oxidative stress and liver dysfunction, namely phenylalanine, tyrosine, bilirubin, and C-reactive protein. This observed negative association strongly suggests that the antioxidant and anti-inflammatory functions of small HDL subclasses may play a protective role in liver failure. The observed elevation of phenylalanine and tyrosine in liver disease has been linked to oxidative stress-induced inhibition of phenylalanine hydroxylase [18]. The observed strong negative correlation suggests a possible direct interaction between the depletion of small HDL subclasses and the development of metabolic dysfunction in this patient population. Similarly, elevated bilirubin levels, indicative of impaired hepatic detoxification, and elevated CRP, a marker of increased systemic inflammation, both showed negative correlations with all HDL subclasses.

Our previous research showed a significant reduction in the activities of key enzymes involved in HDL maturation and metabolism in patients with cirrhosis, including phospholipid transfer protein, lecithin–cholesterol acyltransferase (LCAT), cholesteryl ester transfer protein (CETP), and lipoprotein lipase (LPL) [3]. LCAT facilitates HDL maturation by esterifying free cholesterol, which is then incorporated into the HDL core, while CETP regulates HDL composition by transferring cholesteryl esters between HDL and very-low-density lipoprotein. The marked reduction in the S-HDL and XS-HDL subclasses observed in cirrhosis is probably due to impaired hepatic ApoA-I synthesis combined with these enzymatic deficiencies. Specifically, reduced LCAT activity impairs the conversion of pre-β-1 HDL to α-migrating HDL, limiting cholesterol esterification and resulting in lower cholesterol content within these smaller HDL subclasses. At the same time, reduced CETP activity limits the transfer of cholesteryl esters from HDL, potentially preserving L-HDL levels. During LPL-mediated lipolysis of triglyceride-rich lipoproteins, surface remnants such as phospholipids and apolipoproteins are transferred to HDL [19], supporting HDL remodeling and maturation. However, reduced LDL activity in cirrhosis may interfere with this process and further increase triglyceride levels in HDL by inhibiting the degradation of triglycerides in HDL. These complex changes in HDL subclass distribution and composition warrant further investigation, given the different contributions of each subclass to lipid homeostasis and innate immunity.

The immunomodulatory role of HDL may be particularly important in chronic liver failure. Patients with cirrhosis are highly susceptible to Gram-negative bacterial infections, which cause excessive release of the pro-inflammatory cytokine tumor necrosis factor-alpha (TNF-alpha), exacerbating liver damage [20]. As a result, mortality from septic shock in cirrhotic patients reaches approximately 80% [21], far exceeding the 30% rate observed in individuals without cirrhosis [22]. The diminished quantity and impaired functionality of HDL may substantially contribute to the pathophysiology of systemic inflammation, a key driver in the progression to acute-on-chronic liver failure (ACLF) [23]. HDL exerts a critical protective role against sepsis by neutralizing deleterious bacterial cell wall components, such as lipopolysaccharide (LPS) [10]. Notably, restoring HDL levels and functions with reconstituted HDL significantly attenuated LPS-induced inflammatory pathways in an ex vivo study of patients with advanced chronic liver failure [24]. While our study did not directly assess the risk of decompensation, the observed associations support the hypothesis that HDL subclass profiling may offer early predictive insight into decompensating events and serve as a potential biomarker for risk stratification in chronic liver disease. Future prospective studies are warranted to explore these links.

Lipoprotein receptors play a dual role in viral infection. While they are crucial for lipid metabolism, viruses can exploit these receptors to enter host cells, evade immune responses, and disseminate throughout the body. Specifically, the HDL scavenger receptor B1 (SR-BI) has been shown to facilitate hepatitis B virus (HBV) entry into hepatocytes by interacting with the viral preS1 envelope protein [10]. Notably, HBV infection upregulates SR-BI expression in hepatocytes, potentially enhancing viral replication. However, HDL can competitively bind to SR-BI, blocking viral access and enabling the receptor’s protective function in innate immunity [25]. Understanding the complex relationship between HDL and the immune system may reveal innovative targets for developing new treatments to combat infectious diseases and improve patient outcomes. Implementing NMR spectroscopy in routine clinical laboratories can facilitate high-throughput, standardized measurement of lipoprotein subfractions [26], thereby offering enhanced risk stratification. Furthermore, NMR facilitates comprehensive profiling of HDL subclasses, which have demonstrated potential as biomarkers in conditions such as Alzheimer’s disease [27], acute heart failure [13], and myocardial infarction [12]. Interpreting HDL-ApoA-I in mg/dL offers a protein-centric view of HDL composition rather than a direct count of particles. Reductions in ApoA-I mass within small HDL subclasses may indicate structural and functional impairment, such as reduced antioxidant and anti-inflammatory potential. These changes are consistent with dysfunctional HDL profiles observed in chronic disease states and may have prognostic significance. Although the Bruker IVDr NMR platform used in this study provides standardized and reproducible lipoprotein subclass data, its current application is largely limited to research environments. Broader clinical implementation may be constrained by the need for specialized equipment, technical expertise, and cost considerations. Nevertheless, as NMR technology becomes more automated and cost-effective, its integration into clinical laboratories may become increasingly feasible, particularly for high-throughput risk stratification or biomarker panels, potentially expanding their use from research to clinical diagnostics [28].

Certain limitations of the current study warrant consideration. This study’s observational nature limits our ability to establish causality between HDL subclass changes and mortality. We acknowledge that relying solely on plasma measurements represents a limitation regarding spatial specificity. These systemic markers cannot definitively distinguish the precise contribution of hepatic oxidative stress from oxidative processes originating in other organs or from broader systemic inflammation. Ideally, future studies incorporating direct assessments of intrahepatic oxidative stress through liver tissue biopsies or hepatic-specific imaging/biomarkers would provide a more direct and spatially resolved measure.

The observed depletion of S-HDL and XS-HDL particles in cirrhosis, especially in the compensated stage, points to a potentially modifiable factor in disease progression. Since these HDL subclasses are known for their antioxidant, anti-inflammatory, and cholesterol efflux capabilities, their loss may fuel systemic inflammation and oxidative stress in cirrhotic patients.

Our findings suggest that HDL subclasses may offer superior clinical utility as biomarkers for disease monitoring and risk stratification compared to total HDL cholesterol. This highlights the importance of investigating interventions aimed at preserving or restoring small HDL particles, including ApoA-I mimetic peptides, lifestyle modifications, and novel pharmacological agents, for their potential to ameliorate cirrhosis-related complications. Furthermore, incorporating HDL subclass analysis into clinical practice could significantly improve prognostic assessment beyond established tools like the MELD score. While acknowledging the observational nature of this study precludes causal inference, these insights are pivotal for designing future trials on HDL-targeted therapies in liver disease.

## 5. Conclusions

Taken together, these findings suggest that reductions in specific HDL subclasses may amplify oxidative stress, potentially creating a detrimental feedback loop that exacerbates liver dysfunction and systemic complications in cirrhosis. The robust inverse association between small HDL and these markers of oxidative stress highlights therapeutic potential of strategies targeting restoration of small HDL subclasses. HDL subclass analysis provides robust mortality prediction in compensated cirrhosis, comparable to the MELD score. This indicates the potential for incorporating HDL subclass profiling into clinical practice to enable personalized therapeutic strategies. Such strategies could enhance HDL-mediated functions, including cholesterol efflux, anti-inflammatory, and antioxidant activity, ultimately leading to improved patient outcomes and more targeted anti-inflammatory interventions.

## Figures and Tables

**Figure 1 antioxidants-14-00664-f001:**
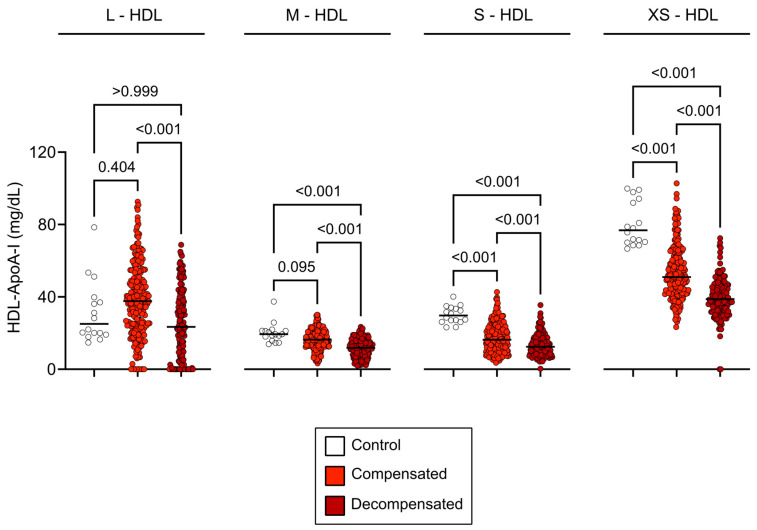
HDL subclass concentrations across study groups. Baseline concentrations of HDL-ApoA-I subclasses in healthy controls, compensated cirrhosis, and decompensated cirrhosis. Concentrations in compensated cirrhosis are represented in light red, decompensated cirrhosis in dark red, and healthy controls in white. The graphs show all data points, with the line representing the median of each data set. Statistical significance was assessed using the Kruskal–Wallis test with Dunn’s multiple comparisons post hoc analysis. Abbreviations: ApoA-I, apolipoprotein A-I; HDL, high-density lipoprotein; L-, large; M-, medium; S-, small; XS-, extra-small.

**Figure 2 antioxidants-14-00664-f002:**
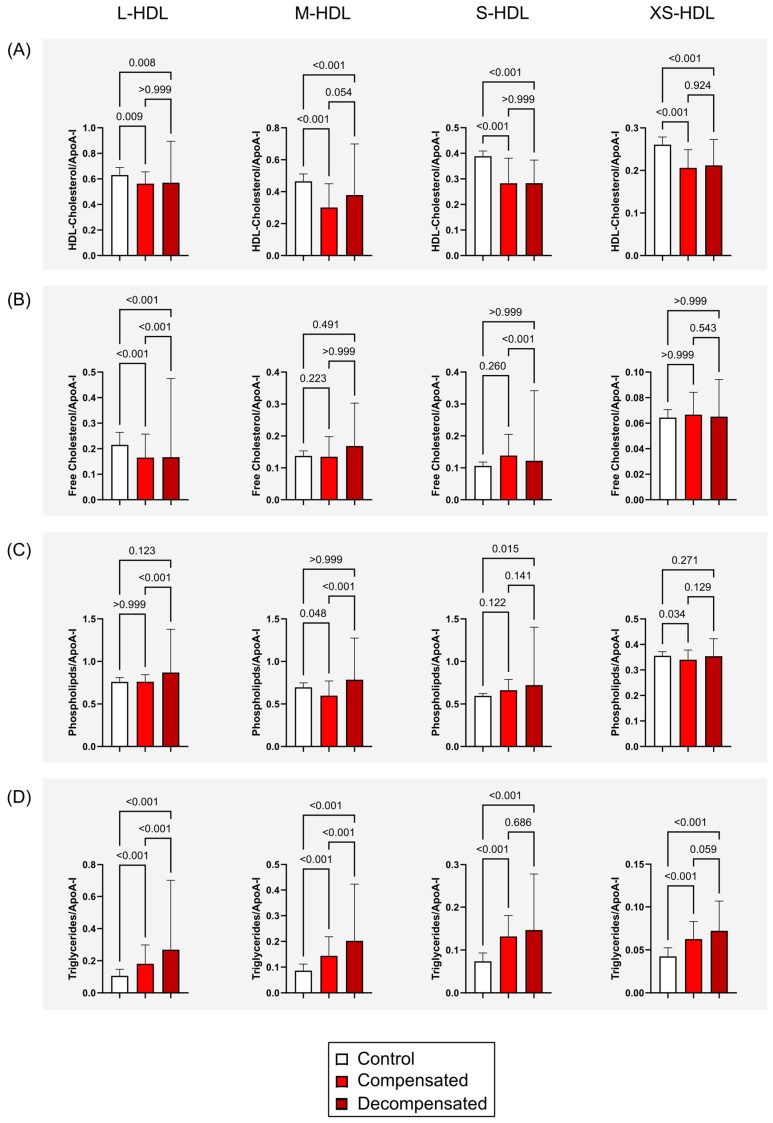
HDL Subclass composition across study groups. HDL lipid subclass measurements were normalized to corresponding ApoA-I subclass levels to assess compositional differences in lipid content across L- to XS-HDL particles. (**A**) HDL-total cholesterol (free and esterified) content in L-HDL to XS-HDL particles in healthy controls and patients with compensated or decompensated cirrhosis. (**B**) HDL-free cholesterol levels across HDL subclasses. (**C**) HDL-phospholipid content in HDL particles. (**D**) HDL-triglyceride levels in L-HDL to XS-HDL particles. Error bars indicate the standard deviation of the mean. Statistical significance was evaluated using the Kruskal–Wallis test followed by Dunn’s multiple comparisons post hoc analysis. Abbreviations: ApoA-I, apolipoprotein A-I; HDL, high-density lipoprotein; L-, large; M-, medium; S-, small; XS-, extra-small.

**Figure 3 antioxidants-14-00664-f003:**
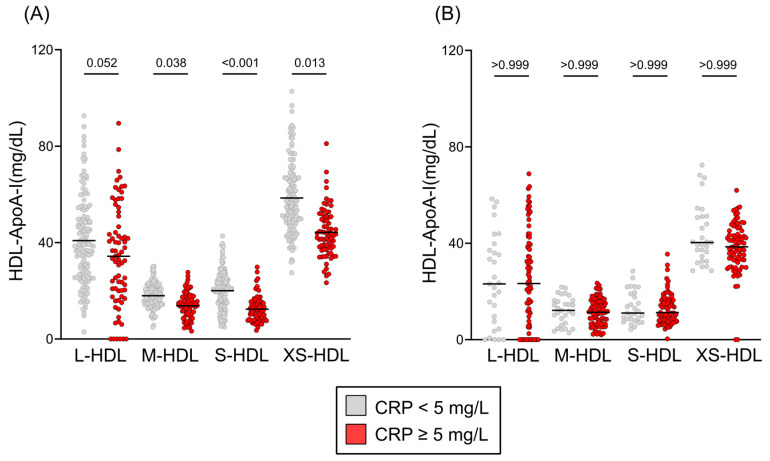
HDL subclass distribution by inflammation status. HDL subclass distributions are presented as Tukey’s box plots, comparing patients with C-reactive protein (CRP) levels <5 mg/L (gray) and ≥5 mg/L (red). Distributions are shown for (**A**) compensated cirrhosis and (**B**) decompensated cirrhosis. Statistical differences were determined using the Kruskal–Wallis test with Dunn’s multiple comparisons post hoc analysis. Abbreviations: ApoA-I, apolipoprotein A-I; HDL, high-density lipoprotein; L-, large; M-, medium; S-, small; XS-, extra-small.

**Figure 4 antioxidants-14-00664-f004:**
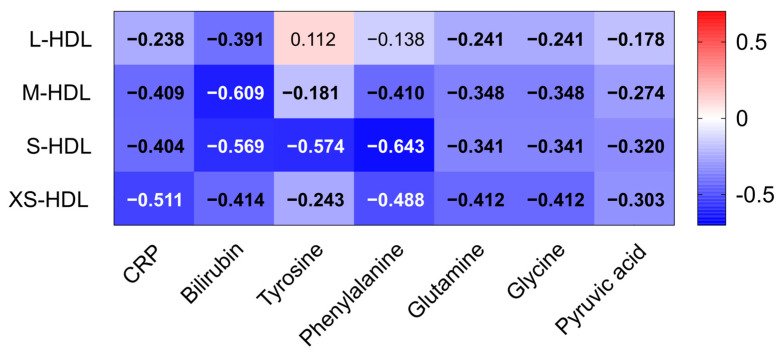
Correlations of HDL subclasses with inflammatory and oxidative markers. Clinical parameters such as CRP and bilirubin as well as NMR-measured amino acid levels of all patients with liver cirrhosis (n = 363) were correlated with HDL subclasses (L-HDL–XS-HDL). Each cell of the heatmap represents a pairwise Spearman’s correlation between the two parameters indicated in the respective row and column. Significant values, after Bonferroni correction, are depicted in bold. Abbreviations: CRP, C-reactive protein; HDL, high-density lipoprotein; L-, large; M-, medium; S-, small; XS-, extra-small.

**Figure 5 antioxidants-14-00664-f005:**
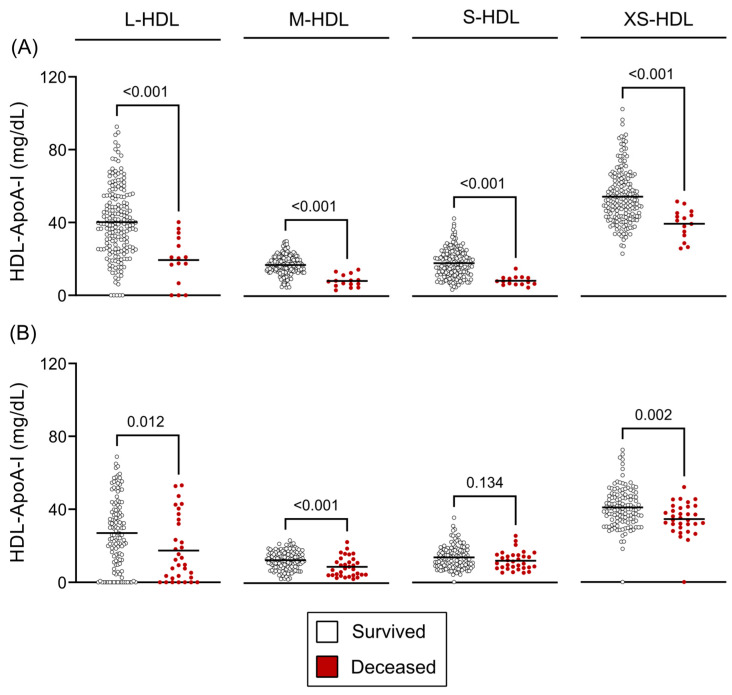
HDL subclass concentrations and survival outcomes. (**A**) Twelve-month survivors versus non-survivors in compensated cirrhosis and (**B**) ninety-day survivors versus non-survivors in decompensated cirrhosis. Statistical significance was assessed using unpaired *t*-tests or Mann–Whitney U tests, as appropriate based on data distribution. Abbreviations: ApoA-I, apolipoprotein A-I; HDL, high-density lipoprotein; L-, large; M-, medium; S-, small; XS-, extra-small.

**Figure 6 antioxidants-14-00664-f006:**
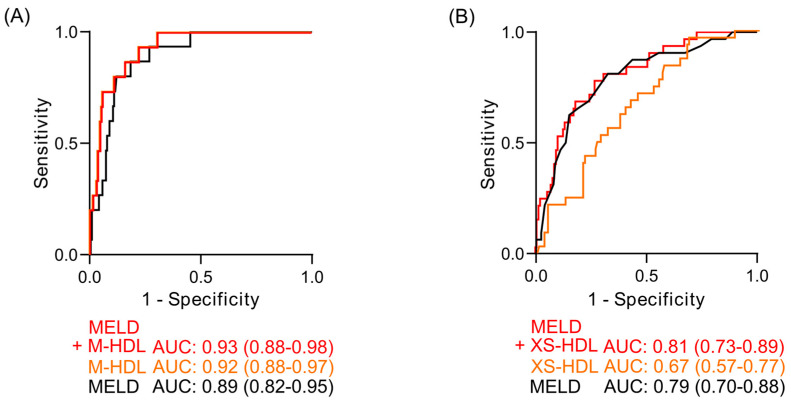
Receiver operating characteristic (ROC) curves for mortality prediction. (**A**) 12-month mortality prediction in compensated cirrhosis (DeLong Test: *p* = 0.612) and (**B**) 90-day mortality prediction in decompensated cirrhosis (DeLong Test: *p* = 0.818). Curves depict the MELD score (black), the HDL subclass significantly associated with mortality (orange), and the combined MELD-HDL subclass model (red). The area under the curve (AUC) is indicated below the graphs, including the 95% confidence interval, which is displayed in brackets. Abbreviations: ApoA-I, apolipoprotein A-I; HDL, high-density lipoprotein; MELD, Model for End-Stage Liver Disease.

**Table 1 antioxidants-14-00664-t001:** Baseline characteristics of the study cohort.

	Compensated Cirrhosis (n = 205)	Decompensated Cirrhosis (n = 158)	*p*-Value
Age (years)	58 (52–63)	58 (51–66)	0.092
Gender (female)	52 (25.4)	61 (38.6)	0.007
MELD Score	11.60 (8.82–16.12)	18.00 (13.00–22.25)	<0.001
Albumin	3.90 (3.30–4.40)	3.00 (2.60–3.40)	<0.001
Bilirubin [mg/dL]	1.39 (0.80–2.87)	2.83 (1.40–6.79)	<0.001
Creatinine [mg/dL]	0.83 (0.72–1.02)	0.90 (0.73–1.55)	0.004
INR	1.29 (1.17–1.49)	1.46 (1.27–1.85)	<0.001
WBC	5.14 (3.90–6.63)	5.70 (4.20–8.25)	0.007
CRP	3.05 (1.20–8.03)	15.50 (4.6–38.10)	<0.001
Total Cholesterol	175.36 (43.45–336.91)	119.79 (92.32–151.17)	<0.001
HDL-Cholesterol	43.37 (0.06–85.13)	28.30 (20.04–40.44)	<0.001
Etiology			<0.001
Alcohol	123 (55.40)	62 (39.20)	
Virus	36 (16.20)	51 (32.30)	
Other	54 (24.3)	28 (17.70)	
90-day mortality	8 (3.90)	32 (20.30)	<0.001
12 months mortality	15 (7.30)	55 (34.80)	<0.001

Participant characteristics are reported as median and interquartile range (Q1–Q3), as well as counts and frequencies (%). Mann–Whitney U Test or Fisher’s Exact Test were used to examine differences in clinical and laboratory characteristics. Abbreviations: CRP, c-reactive protein; INR, international normalized ratio; MELD, model for end-stage liver disease; WBC, white blood cell count.

**Table 2 antioxidants-14-00664-t002:** Multivariable Cox-regression analyses of HDL-related parameters with risk of death in liver cirrhosis.

	Compensated		Decompensated	
Parameter	HR (95% CI) Per 1 SD	*p*-Value	HR (95% CI) Per 1 SD	*p*-Value
Total HDL-ApoA-I	0.43 (0.21–0.87)	**0.019**	0.75 (0.49–1.15)	0.189
L-HDL-ApoA-I	0.49 (0.23–1.04)	0.063	0.83 (0.54–1.28)	0.399
M-HDL-ApoA-I	0.09 (0.03–0.33)	**<0.001**	0.73 (0.46–1.16)	0.178
S-HDL-ApoA-I	0.10 (0.03–0.39)	**<0.001**	0.91 (0.54–1.55)	0.730
XS-HDL-ApoA-I	0.24 (0.10–0.58)	**0.001**	0.46 (0.28–0.78)	**0.004**

The independent association between standardized HDL subclasses and mortality was determined using multivariable Cox regression. Analyses were conducted for 12-month mortality in compensated patients and 90-day mortality in decompensated patients. Hazard ratios (HRs) with 95% confidence intervals (CIs) were computed per 1 standard deviation increase in HDL subclass levels, adjusting for age, sex, and MELD score. Significant findings (*p* < 0.05) are indicated in **bold**. Abbreviations: ApoA-I, apolipoprotein A-I; HDL, high-density lipoprotein; HR, hazard ratio, L-, large; M-, medium; S-, small; XS-, extra small.

## Data Availability

Data are contained within the article and the Appendix A.

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
