# Peer review of "Depletion of Small HDL Subclasses Predicts Poor Survival in Liver Cirrhosis"

_antioxidants, 2025, doi:10.3390/antiox14060664_

Round 1
Reviewer 1 Report
This is a well-executed and clinically relevant study that provides compelling evidence linking reduced levels of small and extra-small HDL subclasses to mortality risk in patients with liver cirrhosis. The use of nuclear magnetic resonance (NMR) spectroscopy for HDL subclass profiling is methodologically robust, and the integration of inflammatory and oxidative stress markers adds mechanistic insight. However, several points require clarification or expansion to strengthen the manuscript.
Major Comments
1.Causality and Clinical Implications
While the authors acknowledge the observational nature of the study, more emphasis should be placed on the limitations this imposes for causal inference. The discussion could benefit from more detailed elaboration on how these findings could inform clinical practice or therapeutic interventions targeting HDL functionality.
2.Control Group Size
The number of healthy controls (n = 16) is relatively small compared to the cirrhosis groups. Although matching was performed, a brief power analysis or justification for control group size would help validate the comparative analysis.
3.Standardization of HDL Subclass Units
The HDL-ApoA-I levels are presented in mg/dL across various HDL subclasses. However, it is not always clear how these were normalized or whether the measurements reflect particle number or mass concentration. The authors should clarify the biological meaning and clinical interpretability of these units.
4.Mortality Endpoint Justification
The study uses 90-day and 12-month mortality endpoints. While appropriate, it would strengthen the clinical applicability if the authors could comment on the potential use of HDL subclass profiling in intermediate endpoints (e.g., infection, hepatic decompensation, or ACLF development).
5.Multivariate Model Composition
In Table 2 (page 10), the Cox regression includes MELD score, age, and sex, but does not account for CRP or bilirubin levels, which were strongly associated with HDL subclasses. The authors should justify this selection or consider including inflammatory markers in sensitivity analyses.
Minor Comments
1.Typographical Details
The abstract uses a nonstandard line break formatting (e.g., lines 1–33). While possibly an artifact of submission formatting, this should be corrected in the final version.
Line 364: “therapeutic promise of strategies aimed at restoring HDL levels” – consider rephrasing to “therapeutic potential of strategies targeting restoration of small HDL subclasses” to match study findings.
2.Figure Legends
Figures 1 and 2: Indicate whether error bars represent standard deviation or standard error of the mean.
Figure 6: The ROC curve AUC comparison would benefit from a statistical comparison (e.g., DeLong test) between MELD and HDL models.
2.Terminology Consistency
Terms such as “S-HDL”, “small HDL”, and “HDL subclasses” are sometimes used interchangeably. It would be helpful to standardize these terms throughout the manuscript for clarity.
3.NMR Platform Generalizability
While Bruker IVDr is a well-established platform, readers would benefit from a brief comment on the feasibility or limitations of implementing this analysis in non-research clinical laboratories.
Author Response
Reviewer 1
Comments 1:This is a well-executed and clinically relevant study that provides compelling evidence linking reduced levels of small and extra-small HDL subclasses to mortality risk in patients with liver cirrhosis. The use of nuclear magnetic resonance (NMR) spectroscopy for HDL subclass profiling is methodologically robust, and the integration of inflammatory and oxidative stress markers adds mechanistic insight. However, several points require clarification or expansion to strengthen the manuscript.
Response 1: We sincerely thank the reviewer for their positive assessment of our study and for recognizing the methodological strengths and clinical relevance of our findings. We appreciate the constructive feedback and have addressed each of the specific comments in detail. Revisions have been made throughout the manuscript to clarify key points, incorporate additional analyses, and enhance the overall interpretation and clinical applicability of the results.
Comments 2: Causality and Clinical Implications While the authors acknowledge the observational nature of the study, more emphasis should be placed on the limitations this imposes for causal inference. The discussion could benefit from more detailed elaboration on how these findings could inform clinical practice or therapeutic interventions targeting HDL functionality.
Response 2: We thank the reviewer for highlighting the need to more explicitly discuss the limitations regarding causal inference and the potential clinical relevance of our findings. We have now expanded the Discussion section to better articulate these points (see lines 408-421).
Comments 3: Control Group Size The number of healthy controls (n = 16) is relatively small compared to the cirrhosis groups. Although matching was performed, a brief power analysis or justification for the control group size would help validate the comparative analysis.
Response 3: While the number of healthy controls is modest, several steps were taken to ensure valid comparative analysis:
- Matching was carefully performed for age, sex, and BMI to reduce confounding, which limited the number of eligible control subjects.
- We now include a brief power analysis in the Methods section. Using standard assumptions (α = 0.05), our analysis showed that with 16 controls and either 205 or 158 patients in the comparison groups, the study had sufficient power (>0.86) to detect large effect sizes (Cohen’s d = 0.8), and over 0.75 power for effect sizes as low as d = 0.7.
- These effect sizes are in line with the magnitude of observed differences in HDL particle composition, supporting the robustness of our comparisons.
We have added the corresponding power estimates to the revised Methods section (see Methods, lines 118-123).
Comments 4: Standardization of HDL Subclass Units. The HDL-ApoA-I levels are presented in mg/dL across various HDL subclasses. However, it is not always clear how these were normalized or whether the measurements reflect particle number or mass concentration. The authors should clarify the biological meaning and clinical interpretability of these units.
Response 4: According to the reviewer’s suggestion, we now explicitly state in the revised manuscript that:
- HDL-ApoA-I levels (mg/dL) represent the mass concentration of apolipoprotein A-I within each HDL subclass, as quantified by nuclear magnetic resonance (NMR) spectroscopy.
- These values do not reflect direct particle counts, but instead indicate the amount of ApoA-I protein associated with each subclass in plasma.
- Since ApoA-I is the main structural protein of HDL particles, its distribution across subclasses provides insight into HDL remodeling and function in disease.
- Clinically, changes in HDL-ApoA-I mass, particularly within small HDL particles, may reflect impaired HDL metabolism and function, which has been linked to adverse outcomes in cardiovascular and liver diseases[1].
These clarifications have been added to the Methods section (lines 102-111) and expanded upon in the Discussion (lines 390-402).
Comments 5: Mortality Endpoint Justification The study uses 90-day and 12-month mortality endpoints. While appropriate, it would strengthen the clinical applicability if the authors could comment on the potential use of HDL subclass profiling in intermediate endpoints (e.g., infection, hepatic decompensation, or ACLF development).
Response 5: We appreciate the reviewer's excellent suggestion to discuss the potential for HDL subclass profiling in relation to intermediate clinical endpoints. We've now expanded the Discussion section to highlight how HDL functionality plays a role in processes that can lead to adverse outcomes like hepatic decompensation, infection, and ACLF development. HDL particles, especially the small and extra-small subclasses, are known for their crucial anti-inflammatory and immunomodulatory effects [2]. When these particles are depleted, it can impair the body's natural defenses, increasing susceptibility to infection and systemic inflammation. These are both central to the progression of ACLF and decompensation events [3]. While our study specifically focused on 90-day and 12-month mortality, the strong associations we observed between altered HDL subclasses and markers of oxidative stress and liver dysfunction strongly suggest that HDL profiling could also be valuable for predicting clinically significant non-fatal events. We've incorporated this discussion into the revised manuscript (lines 366-370).
Comments 6: Multivariate Model Composition. In Table 2 (page 10), the Cox regression includes MELD score, age, and sex, but does not account for CRP or bilirubin levels, which were strongly associated with HDL subclasses. The authors should justify this selection or consider including inflammatory markers in sensitivity analyses.
Response 6: We thank the reviewer for this comment. Bilirubin levels are already incorporated in the analysis as they are part of the MELD score (MELD = 3.78 × ln (bilirubin) + 11.2 × ln (INR) + 9.57 × ln (creatinine) + 6.43). According to the reviewer’s suggestion, we have performed an additional analysis by incorporating CRP (log-transformed) levels in the multivariable Cox regression model.
Further adjustment for CRP levels revealed nuanced insights. For compensated patients, the protective association between HDL subclasses and three-month mortality weakened, and the effect of XS-HDL particles lost statistical significance. Conversely, for decompensated patients, the association with XS-HDL particles persisted and strengthened. These additional analyses have been added to the supplemental materials (see Supplemental Table S1, and we have updated the Discussion: lines 312-319).
|
Compensated |
Decompensated |
||
Parameter |
HR (95 % CI) Per 1 SD |
p-value |
HR (95 % CI) Per 1 SD |
p-value |
Total HDL-ApoA-I |
0.74 (0.29-1.91) |
0.541 |
0.65 (0.33-1.27) |
0.209 |
L-HDL-ApoA-I |
0.89 (0.38-2.06) |
0.780 |
0.84 (0.49-1.46) |
0.538 |
M-HDL-ApoA-I |
0.17 (0.04-0.71) |
0.015 |
0.64 (0.33-1.22) |
0.171 |
S-HDL-ApoA-I |
0.04 (0.01-0.27) |
0.001 |
0.95 (0.42-2.11) |
0.891 |
XS-HDL-ApoA-I |
0.55 (0.20-1.52) |
0.246 |
0.33 (0.17-0.66) |
0.001 |
Comments 7: Typographical Details
The abstract uses a nonstandard line break formatting (e.g., lines 1–33). While possibly an artifact of submission formatting, this should be corrected in the final version.
Line 364: “therapeutic promise of strategies aimed at restoring HDL levels” – consider rephrasing to “therapeutic potential of strategies targeting restoration of small HDL subclasses” to match study findings.
Response 7: We thank the reviewer for pointing out these issues. The line breaks in the abstract have now been corrected in the revised version. According to the reviewer’s suggestion, we have rephrased the sentence on lines 366 – 370.
Comments 7: Figure Legends
Figures 1 and 2: Indicate whether error bars represent standard deviation or standard error of the mean.
Figure 6: The ROC curve AUC comparison would benefit from a statistical comparison (e.g., DeLong test) between MELD and HDL models.
Response 7: We thank the reviewer for these helpful suggestions. We have now clarified in the figure legends of Figures 1 and 2 that the error bars represent either the median with all data points or the standard deviation of the mean.
Regarding Figure 6, we performed a DeLong test to statistically compare the AUCs of the MELD score and HDL-based models. The test did not show a statistically significant difference between the models (p = 0.6121, p = 0.8182). For this reason, we did not include the results directly in the figure, but we mention this in the revised Results section for full transparency (lines 288 - 292).
Comments 8: Terminology Consistency Terms such as “S-HDL”, “small HDL”, and “HDL subclasses” are sometimes used interchangeably. It would be helpful to standardize these terms throughout the manuscript for clarity.
Response 8: According to the reviewer’s suggestion, we have carefully reviewed the manuscript and standardized the terminology throughout.
Comments 9: NMR Platform Generalizability While Bruker IVDr is a well-established platform, readers would benefit from a brief comment on the feasibility or limitations of implementing this analysis in non-research clinical laboratories.
Response 9: We thank the reviewer for this thoughtful comment. We have added a brief note in the Discussion to address the feasibility of implementing NMR-based HDL subclass analysis in clinical settings. While the Bruker IVDr platform is standardized and validated for lipoprotein profiling, its current use is primarily confined to specialized research laboratories due to cost, infrastructure, and personnel requirements. However, ongoing advances in NMR automation and data integration are gradually improving the accessibility and scalability of this technology. We highlight these considerations in the revised Discussion section (lines 377 - 384).
References
[1] E.P.C. van der Vorst, High-Density Lipoproteins and Apolipoprotein A1, in: U. Hoeger, J.R. Harris (Eds.), Vertebrate and Invertebrate Respiratory Proteins, Lipoproteins and Other Body Fluid Proteins, Springer International Publishing, Cham, 2020: pp. 399–420. https://doi.org/10.1007/978-3-030-41769-7_16.
[2] A. Kontush, M.J. Chapman, Antiatherogenic small, dense HDL—guardian angel of the arterial wall?, Nat Rev Cardiol 3 (2006) 144–153. https://doi.org/10.1038/ncpcardio0500.
[3] J. Clària, R.E. Stauber, M.J. Coenraad, R. Moreau, R. Jalan, M. Pavesi, À. Amorós, E. Titos, J. Alcaraz‐Quiles, K. Oettl, M. Morales‐Ruiz, P. Angeli, M. Domenicali, C. Alessandria, A. Gerbes, J. Wendon, F. Nevens, J. Trebicka, W. Laleman, F. Saliba, T.M. Welzel, A. Albillos, T. Gustot, D. Benten, F. Durand, P. Ginès, M. Bernardi, V. Arroyo, for the C.S.I. of the E.-C.C. and the E.F. for the S. of C.L. Failure (EF‐CLIF), Systemic inflammation in decompensated cirrhosis: Characterization and role in acute‐on‐chronic liver failure, Hepatology 64 (2016) 1249. https://doi.org/10.1002/hep.28740.
Reviewer 2 Report
This investigation demonstrates that depletion of small and extra-small HDL particles is closely associated to liver cirrhosis, and linked to oxidative stress and mortality in patients with compensated cirrhosis. Some minor issues still need to be addressed.
- The authors should discuss whether the oxidative stress found in the patients' plasma reflects an oxidative state in the liver. That is, can the results found in plasma be extrapolated to liver tissue, or are they independent of each other? Is this a limitation of the research?
- Authors did not include the number of the protocol approved by the institutional ethical committee for the use of human plasma samples.
- Tables of the manuscript should be presented in scientific format. So, the current format should be corrected. Authors should check already published articles to correct them.
- Real p value (the calculated one) must be included throughout description of result section immediately after where authors cite a comparison; i.e., p = 0.001, but not only the minimum accepted p value (p ≥ 0.05).
Please, see major comments section
Author Response
Comments 1: This investigation demonstrates that depletion of small and extra-small HDL particles is closely associated to liver cirrhosis, and linked to oxidative stress and mortality in patients with compensated cirrhosis. Some minor issues still need to be addressed.
Response 1: We thank the reviewer for the positive evaluation of our work and for highlighting the relevance of our findings. We have carefully addressed the minor issues raised and provide detailed responses to each point below.
Comments 2: The authors should discuss whether the oxidative stress found in the patients' plasma reflects an oxidative state in the liver. That is, can the results found in plasma be extrapolated to liver tissue, or are they independent of each other? Is this a limitation of the research?
Response 2: We thank the reviewer for this insightful comment.
We acknowledge that relying solely on plasma measurements represents a limitation regarding spatial specificity. These systemic markers cannot definitively distinguish the precise contribution of hepatic oxidative stress from oxidative processes originating in other organs or from broader systemic inflammation. Ideally, future studies incorporating direct assessments of intrahepatic oxidative stress through liver tissue biopsies or hepatic-specific imaging/biomarkers would provide a more direct and spatially resolved measure. We have expanded the limitation section of the revised manuscript (lines 401 - 407).
Comments 3: Authors did not include the number of the protocol approved by the institutional ethical committee for the use of human plasma samples.
Response 3: We appreciate the reviewer's observation. The protocol numbers approved by our institutional ethics committee (23-056 ex 10/11, 23-096 ex 10/11, 23-258 ex 10/11) have now been moved from the Institutional Review Board Statement section to the Materials and Methods section in the revised manuscript for improved visibility.
Comments 4: Tables of the manuscript should be presented in scientific format. So, the current format should be corrected. Authors should check already published articles to correct them.
Response 4: We appreciate the reviewer's helpful comment regarding table formatting. We've revised all tables in the manuscript to conform to the standard scientific format, drawing guidance from recently published articles in Antioxidants.
Comments 5: Real p value (the calculated one) must be included throughout description of result section immediately after where authors cite a comparison; i.e., p = 0.001, but not only the minimum accepted p value (p ≥ 0.05).
Response 5: According to the reviewer’s suggestion, we have revised the Results section to include the exact p-values for all statistical comparisons, as recommended. Threshold expressions such as p < 0.05 or p > 0.05 have been replaced with the actual computed values throughout the manuscript.
Round 2
Reviewer 2 Report
All my suggestions were properly addressed.
All my suggestions were properly addressed.